# Innovative Methodologies in a Pandemic: The VESS Model

**María Helena Romero-Esquinas, Juan Manuel Muñoz-González \*** 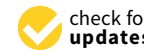 **and María Dolores Hidalgo-Ariza**

Departament of Education, University of Córdoba, 14071 Córdoba, Spain; m32roesm@uco.es (M.H.R.-E.); lola.hidalgo@uco.es (M.D.H.-A.)

\* Correspondence: juan.manuel@uco.es; Tel.: +34-957212561

**Abstract:** The VESS (Meaningful Life with Balance and Wisdom) model is considered to be a learning method based on the construction of knowledge through critical and visible thinking, with a neuroeducational base. The aim of the present work is to conduct a psychometric study of a measurement scale about the learning of the VESS model created for teachers-in-training. This article presents two survey-based descriptive studies conducted at the University of Cordoba (Spain). The data were subjected to descriptive, correlational, reliability and validity analysis through exploratory and confirmatory analyses, respectively. It is an instrument with high goodness-of-fit indices and suitable validity and reliability values. This instrument is applicable to similar study contexts.

**Keywords:** validation; questionnaire; VESS model; methodology; socrative; cooperative learning; visible thinking

---

## 1. Introduction

The global pandemic situation we are currently living in due to a coronavirus has compelled numerous governments to decree a state of alarm to decrease the propagation of the virus. We are facing high rates of infection due to this new coronavirus, which was named Covid-19 by the World Health Organization (WHO) [1]. This state of alarm and subsequent confinement measures have resulted in the paralysis of every social and economic activity in the world, to a greater or lesser degree.

The impact of the coronavirus in Spain is evident. As such, is the case that, from 14 March 2020, schools had to close their doors and teachers had to perform their teaching duties from home. This situation has meant that society in general, and the education community in particular, have had to organize and re-invent themselves based on the principle of autonomy of the centers [2], which for the teachers has been a great challenge and signified a period for thinking, re-organizing and re-defining education.

The author of [3] claims that there are cycles in all areas that make up humanity. Therefore, it is relatively easy to anticipate what is yet to come, and this gives us the option to prosper, taking advantage of weaknesses and turning them into strengths. In this sense, the crises, either due to war, or economic or health-related, accelerate the positive as well as the negative events [4].

Thus, the health crisis caused by covid-19 could have accelerated the irruption and consolidation of new methodologies and technologies in the classroom, as teachers have had to make a great effort and dare to unlearn to learn, or "learn, unlearn, and re-learn", as nuanced by [5], to give a Copernican twist to education.

Unexpectedly, we are moving towards a change in educational philosophy during a full pandemic. The teacher's role, for the first time and almost in a mandatory manner, has been called into question.

The important issue is not only to achieve a curricular level, but to also provide for the emotional well-being of students, as well as their families.

The Education 2030 project identifies "transforming competencies" to change society from its foundations, which is education. These are: creating new value— "adaptability, creativity, curiosity and open mind", reconciling tensions and dilemmas— "learning to think and act in a more comprehensive manner, taking into account the interconnections and interrelations between contradictory ideas, logics and positions, from both short- and long-term perspectives", and taking responsibility— "with morality, intelligence and maturity, with which a person can reflect upon and evaluate one's actions" [6].

In this equation for the development of sustainable education, it is important to reflect on the real function of the school as an institution and the rest of the education community as part of it.

Thus, as indicated by [7], the important issue related to everything that has occurred is to see education as "a holistic phenomenon determined by a set of independent factors . . . so that it brings the opportunity to develop skills and provide guarantees for the rights to education" (p. 3).

In this sense, the VESS (Meaningful Life with Balance and Wisdom) model, born in Harvard, created and developed by "Edu1st", expects educational and social transformation through thinking as the main pedagogic strategy, which allows the students to acquire new knowledge in an equilibrated manner, with meaning and wisdom.

In addition, it underlines the need for emotional restraint of the teaching team, parents, and students, as it implies a fundamental part of this social and educational stage we are currently living in. In this manner, we will be providing the human focus needed for fomenting the comprehensive learning of students. For this, from a teaching point of view, guided, creative, critical, empathic and significant learning should be promoted that complements the information provided by the teacher in a telematics manner [8].

On their part, ref. [9] sustain that emotional education creates a symbiosis between thinking, emotion and action in students, which allows them to face everyday problems without damaging their self-esteem.

Together with this, the communication styles that are established are determining factors in the teaching–learning process. According to [10], the achievement of lessons are reached in an effective and significant manner when there is two-way communication that is open to listening and empathic observation. For this, the priority of the school is not to teach concepts to the students so that they memorize them. Learning should no longer be thought of as external to the subject [11], but it is very important to integrate different critical, creative and reflective thinking skills into learning and the curriculum [12].

In this sense, what is most important is to create links between school and life, having in mind the cultural and social context to discern what lessons are more pertinent and useful in a given moment, and similarly, learning to create significant spaces for living well [13].

In this respect, ref. [14] asserts that the learning process requires concentration, attention, connection and relation of what was learned with what was already in one's mind. Nevertheless, apart from the basic functions, learning also requires "the mental inhibition of all of those thoughts or emotions that constantly assault our mind and distract it" (p. 114). In the context of the confinement we have lived through, this last part is especially relevant. So, ref. [15] identify some characteristics that distance-learning must have. These are: connectivity, student-centeredness, unboundedness, community, exploration, shared knowledge, multisensory experience, and authenticity. Along this line, the VESS model proposes learning by creating connections through experiential, critical and visible learning for achieving a global comprehension of the world and a complete personal development.

Now, more than ever, students are being educated in more immediate socializing environments. In addition, ref. [16] assert that this type of education is deeper and more real as compared to formal education. Nevertheless, the latter is more complete and safer. For this, it is important that teachers act as "hinges" to combine formal learning with the day-to-day experiences of the student. In addition, it considers that each individual cannot construct knowledge in isolation without considering the rest

of the people or that which surrounds him or her. For this reason, "cooperative intelligence" is now being discussed. From the neuroeducational point of view, different theories exist that sustain the theory of social or vicarious learning, founded on imitation and autonomous discovery. The mirror neurons discovered by Rizzolatti (1996), have a great weight in learning. The report by [17] shows the results from cognitive neuroscience related to social learning, underlining that these neurons play a fundamental role in comprehension through the actions performed by others, as well as learning through imitation and observation.

This cooperative manner, which should be prioritized in the teaching–learning process, is greatly determined by the language of thought. Project Zero, from Harvard University, describes this concept, found in [18].

In this sense, ref. [19] sustains that language foments thinking and learning, as it makes it explicit and visible. In this sense, the technology for learning and knowledge (TLK) makes visible the curriculum that will be addressed at school at a given moment in time. Furthermore, the information and communication technologies (ICT) bring together knowledge, language and communication, making possible the complete set of online learning in a manner that is two-way and combined.

In addition, the ICT-TLK have a "hidden curriculum", as defined and elaborated upon by [20] and [21], respectively, and represent a learning space that is both explicit and implicit, through images that foment critical, cooperative and dialogic thinking, which stimulates the students. The use of technologies has radically irrupted into education. In such a way that the processes that shape teaching and learning have been modified, and this has also been observed with the process of assessment.

Some authors, ref. [22] define evaluation around comprehension. Quite literally, they sustain that "it is a process through which the students gain understanding about their own competencies and make progress, as well as a process through which they are scored." (p. 128). The Socrative software, as an assessment tool, makes possible instant feedback that is motivating, cooperative, and which facilitates the acquisition of knowledge during the assessment process. Nevertheless, we should be aware that during the present pandemic, assessment should be viewed from a "self-assessment" perspective by the students and also the teachers. The scenario that this health crisis shows us compels teachers to "trim the curriculum" to guarantee justice, equity, equality and efficiency in teaching, and online or distance assessment. Likewise, the personal situation of the student, and the family's as well, should be taken into consideration [23].

The family, in this sense, has great importance in the teaching–learning process of the student, and even more so considering the exceptional time we are currently living in. For this reason, the authors of [24] affirm that the household environment has an influence on the construction of personality, attitudes and values of the student.

## 2. Materials and Methods

This work is based on the process of validation and analysis of the technical characteristics of the instrument created ad hoc, named Questionnaire on the perceptions of the VESS model in Higher Education. In this respect, other research studies which have analyzed constructs on the perception of the VESS model in higher education are lacking. Thus, the present work intends to create a positive impact for social and educational advancement, through the implementation of this instrument which considers the expectations of the use of the VESS method, cooperative learning and the Socrative software for the evaluation of the learning process.

The methodology selected forms part of a cross-sectional survey-based research study, given the numerical and reliable nature of the data collected, and due to the research strategy utilized, which is deductive and structured. Within this frame, two studies were conducted. The first was a pilot study which utilized about half of the sample, and was therefore exploratory in nature; the second utilized the total research sample and was therefore a confirmatory study [25].

## 2.1. Sampling and Participants

To select the sample, a non-probabilistic, convenience sampling method was utilized [26], given that the implementation of the methodology, as well as the application of the questionnaire, could only be conducted with the students to whom the study's director taught during the 2018–2019 academic year.

Two studies were conducted, which corresponded to the exploratory factorial analysis (EFA), taking into account approximately half of the total sample: 128 students, and the confirmatory factorial analysis (CFA) comprised by the entire sample: 231 participants.

Table 1 shows the description of the sample, according to the two studies that were previously mentioned.

**Table 1.** Characteristics of the EFA (study 1) and CFA (study 2) sample.

| Variables | Study 1: N = 128 | | Study 2: N = 231 | |
|---|---|---|---|---|
| | Male | Female | Male | Female |
| **Gender** | 17.97% | 82.03% | 19.05% | 80.95% |
| **Studies** | | | | |
| Early Childhood Education Degree | 4.11% | 95.89% | 3.94% | 96.06% |
| Primary Education Degree | 39.22% | 60.78% | 39.56% | 60.44% |
| Masters in Inclusive Education Degree | 0% | 100% | 23.08% | 76.92% |
| **Age range** | | | | |
| 17–19 years old | 16.67% | 83.33% | 22.37% | 77.63% |
| 20–22 years old | 22% | 78% | 18.6% | 81.4% |
| 23–25 years old | 9.09% | 90.91% | 11.11% | 88.89% |
| More than 26 years old | 21.43% | 78.57% | 25% | 75% |
| **School** | | | | |
| Public | 17.82% | 82.18% | 18.24% | 81.76% |
| Concerted | 18.52% | 81.48% | 18.97% | 81.03% |
| Private | 0% | 0% | 66.67% | 33.33% |

EFA: Exploratory Factor Analysis; CFA: Confirmatory Factor Analysis.

## 2.2. Data Collection Instrument

The tool utilized to conduct this study was a survey, named Questionnaire on the perceptions of the VESS model in Higher Education. This was an ad hoc, online questionnaire with close-ended questions, poly-thematic in character and with a Likert-type response scale with five possible answers, ranging from total disagreement (1) to total agreement (5) in the dimensions "Perspectives of the student on the VESS model", "Cooperative learning", and "Socrative". In addition, it included independent variables related to academic (degree studied) and sociodemographic (age, sex, type of school where they studied, and current studies) characteristics.

It contained a set of 16 items written as statements and structured into three dimensions:

–Perspectives of the student on the VESS model: this refers to the characteristics, perceptions, and expectations of Higher Education students on the VESS model. It comprises 7 items.

–Cooperative learning: Composed of 5 items, it alludes to the perception of the students enrolled in the Primary Education, Early Childhood Bachelors, and Masters in Inclusive Education degrees at the University of Cordoba (Spain), about working in teams and cooperative learning as the basis of this model.

–Socrative: This last dimension is shaped by 4 items which refer to the perception on the use of this digital tool for assessment and self-assessment.

In summary, the dimensions and items included in the questionnaire are the following (see Table 2):

**Table 2.** Dimensions and items of the questionnaire.

| Dimension | Items |
|---|---|
| Factor 1: Perspectives of the student on the VESS model | 1. The VESS model considers diversity.<br>2. The VESS model foments creativity.<br>3. In Early Childhood Education, it makes sense to introduce these methodologies into the classroom.<br>4. Through the VESS model, it is possible to work on the contents from a holistic point of view.<br>5. Establishing thinking routines is necessary for guiding the students in the learning processes.<br>6. The VESS model foments learning through the externalization of thought.<br>7. The VESS model allows the organization of the content in the minds of children. |
| Factor 2: Cooperative learning | 8. I consider that cooperative learning is a good method to use for developing my social competencies.<br>9. Cooperative learning makes me feel as an active part of my own learning process.<br>10. The performance of the group improves if the activities planned by the teachers require reflection.<br>11. Working cooperatively is a way to better understand the contents.<br>12. Cooperative learning allows me to reach an agreement between different opinions. |
| Factor 3. Socrative | 13. Socrative allows digital development in the classrooms.<br>14. Socrative allows the participation of the students in real time.<br>15. Socrative involves all the students in an active manner.<br>16. Socrative foments the creation of playful learning. |

VESS: Balanced Life with Meaning and Knowledge (Vida Equilibrada con Sentido y Sabiduría).

*2.3. Data Analysis*

This questionnaire was designed and created online with the Google Forms software. Thus, its completion was fast and effective. The process was closely monitored by the researchers involved in the study, so that any difficulty during the completion process or comprehension was detected and solved.

Once the instrument was created, the in-person data collection was conducted. Before this, some time was spent for learning about the model. Each group involved was provided with an explanation on the foundation on which the VESS model was based, and the students worked with it, designing didactic proposals. Afterwards, freely and without a limit in time, the participants answered the questionnaire. In the first study, a pilot test of the instrument was performed to adapt it and contextualize it, taking into consideration the object of study population utilized in the research study. This pilot session allowed for pointing out if the items were pertinent as related to their understanding and ambiguity, and also allowed verification of their discrimination indices and the analysis of the factorial structure of the instrument.

After collecting the information, the content of the items was analyzed. In this case, there was no need to normalize the sample, as appropriate values were found (K-S, $p > 0.5$) [27,28] through an exploratory factorial analysis, using Pearson's correlation matrices, along with the "Optimal implementation of parallel analysis" method [29] to determine the number of factors, and the method for the extraction of common factors, "Robust maximum likelihood" with a "Promin" rotation

criteria [30]. Next, its internal consistency was analyzed [31]. This analysis was performed with the SPSS 25 statistical package, and the Factor Analysis (10.8.04) program.

After analyzing the characteristics of the first study, a second confirmatory study was developed, with the entire study sample. For collecting the data, the same guidelines as study 1 were utilized.

As in the previous study, the normalization of the sample was not needed, as appropriate values were found (K-S, $p > 0.5$) [27,28]. Through the confirmatory factor analysis, and with the help of the AMOS 25 software program, structural equation models were utilized and the fit of the model was evaluated with the following statistical tests: $\chi^2$ test/degrees of freedom [32], comparative fit index (CFI), incremental fit index (IFI), normed fit index (NFI), the Tucker–Lewis index (TLI) [33,34], the root mean square (RMR), the root mean square error of approximation (RMSEA) [35], and the expected cross validation index (ECVI).

Next, the convergent and discriminant validities of the instrument were analyzed with the previously-mentioned software, taking into account the indices recommended in the literature: Composite Reliability (CR), Average Variance Extracted (AVE), Maximum Shared Variance (MSV), and the Maximum Reliability Coefficient H (MaxR(H)). Next, the reliability of both the instrument and its parts was analyzed, as well as each of its dimensions, through an internal consistency study.

Lastly, a correlational study of all the different dimensions that comprised the questionnaire was performed.

## 3. Results

### 3.1. Exploratory Factor Analysis

Study 1:

To verify the validity of the construct, an exploratory factorial analysis (EFA) was performed through the use of the Optimal Implementation of Parallel Analysis method [29], along with the process of extraction of common factors, "Robust Maximum Likelihood" (RML), having in mind the "Weighted Oblimin" rotation method [30] whose Kaiser–Meyer–Olkin (KMO) index was 0.94, Bartlett's sphericity test $p = 0.000$ and RMSR = 0.0379, and was thus deemed appropriate for the model [36].

Once the criteria were verified, the EFA was applied to the original version of the questionnaire (composed of 16 items and three dimensions), with a total of three factors to be extracted.

The results of the extraction of principal components showed the existence of three factors, where the total explained variance was 68.33%. The commonalities, on their part, were found between 0.308 in item 5, and 0.997 in item 8, with all the variables found to be above 0.3 [37]. By analyzing the matrix of rotated factors and the value of each item according to factor (see Table 3), the presence of a high correspondence with the different dimensions considered in the study was observed in the entire set of items, with loads higher than 0.3. Thus, an adequate equilibrium existed between the components of the instrument, which represented the theoretical concept.

The measurement of the reliability of the instrument was conducted through an internal consistency focus [38], with a value for Cronbach's Alpha, considering the instrument as a whole, of 0.894, and specifically for the three factors studied, of $\alpha = 0.834$ for factor 1; $\alpha = 0.738$ for factor 2, and $\alpha = 0.753$ for factor 3. In all the cases, the results obtained indicated a high reliability.

**Table 3.** Matrix of rotated factors.

| Variable | F 1 | F 2 | F 3 |
|---|---|---|---|
| V 1 | | | 0.431 |
| V 2 | | 0.391 | |
| V 3 | | | 0.498 |
| V 4 | 0.359 | | |
| V 5 | 0.308 | | |
| V 6 | 0.336 | | |
| V 7 | 0.446 | | |
| V 8 | 0.997 | | |
| V 9 | 0.529 | | |
| V 10 | | 0.622 | |
| V 11 | | 0.614 | |
| V 12 | | 0.867 | |
| V 13 | | | 0.573 |
| V 14 | | | 0.730 |
| V 15 | | | 0.662 |
| V 16 | | | 0.575 |

*3.2. Confirmatory Factor Analysis*

Study 2:

To confirm the model obtained through the EFA, a Confirmatory Factorial Analysis was performed (from here on, CFA), with the Maximum Likelihood as the estimation method. The results showed that, except for four items, the rest obtained standardized factorial loads with values higher than 0.4. Thus, 12 of the variables integrated in the scale were kept (Figure 1).

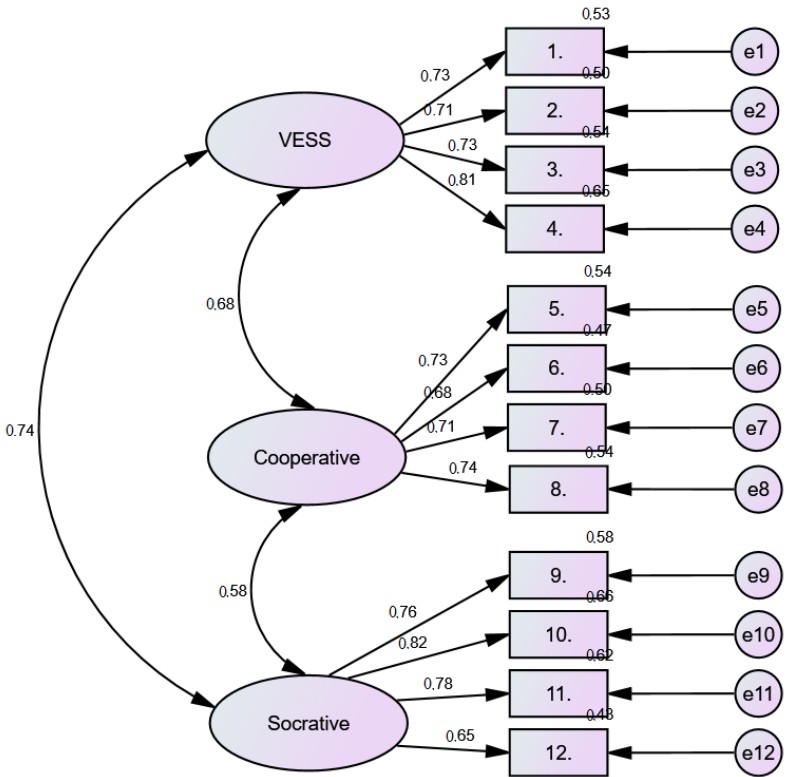

**Figure 1.** Three factor model (CFA).

To evaluate the fit of the model identified, on the one hand, the following tests were considered: $\chi^2$ test/degrees of freedom, the comparative fit index (CFI), normed fit index (NFI), incremental fit index (IFI), the Tucker–Lewis index (TLI), the root mean square (RMR), the root mean square error of approximation (RMSEA), and the expected cross validation index (ECVI). On the other hand, the convergent and discriminant validities of the instrument were verified, taking into account the indices recommended in the literature: Composite Reliability (CR), Average Variance Extracted (AVE), Maximum Shared Variance (MSV), and the Maximum Reliability Coefficient H (MaxR(H)), with the results shown in Tables 4 and 5.

**Table 4.** Fit indices of the model.

|  | $\chi^2$ | df | p | $X^2$/df | CFI | IFI | NFI | NNFI (TLI) | RMSEA | ECVI |
|---|---|---|---|---|---|---|---|---|---|---|
| Values | 69.6 | 49 | 0.028 | 1.42 | 0.982 | 0.983 | 0.944 | 0.976 | 0.043 | 0.676 |

**Table 5.** Validity coefficients of the 3-factor model.

|  | CR | AVE | MSV | MaxR(H) | Socrative | VESS | Cooperative |
|---|---|---|---|---|---|---|---|
| **VESS** | 0.842 | 0.573 | 0.551 | 0.851 | **0.757** | | |
| **Cooperative learning** | 0.833 | 0.556 | 0.551 | 0.916 | 0.742 | **0.745** | |
| **Socrative** | 0.808 | 0.512 | 0.461 | 0.938 | 0.575 | 0.679 | **0.716** |

Lastly, the internal consistency was analyzed, considering the instrument as a whole, as well as its dimensions, with the following results obtained.

As shown in Table 6, all the factors of the instrument obtained coefficients that were higher than 0.700, an index that is considered acceptable [39]. These results allowed us to confirm that the instrument had a high internal consistency as a whole and according to its dimensions.

**Table 6.** Internal consistency of the instrument.

| Dimensions | Reliability |
|---|---|
| Factor 1: Perspectives of the students on the VESS model | $\alpha$ = 0.832 (n = 4) |
| Factor 2: Cooperative learning | $\alpha$ = 0.782 (n = 4) |
| Factor 3. Socrative | $\alpha$ = 0.836 (n = 4) |
| Total | $\alpha$ = 0.892 (n = 12) |

### 3.3. Descriptive Analysis

To answer the objectives of the study, a descriptive analysis of the dimensions of the questionnaire was conducted. This analysis showed a trend which provided evidence, to a greater or lower degree, of the relationships between the different dimensions. In this sense, it was concluded that there was a trend towards agreement with respect to the future use of the VESS model, with means close to 5 points, as shown on Table 7.

**Table 7.** Descriptive statistics according to dimension.

| Dimensions | M. | Me. | DT | Min. | Max. |
|---|---|---|---|---|---|
| Perspectives of the students on the VESS model | 4.32 | 4.25 | 0.56 | 2.25 | 5.00 |
| Cooperative learning | 4.32 | 4.5 | 0.58 | 2.5 | 5.00 |
| Socrative | 4.42 | 4.5 | 0.58 | 2.75 | 5.00 |

M.: Mean; Me.: Median; DT: Standard Deviation; Min.: Minimum; Max: Maximum.

In this table, it is observed that the dimension "Socrative" obtained the highest mean (4.42), which reveals that the perception of the students surveyed on the use of this digital tool in the classrooms is positive. As for the percentages, this dimension showed that option 5 was the most-often repeated (33.3%).

Likewise, dimension 1, which alludes to the student's perspective on the VESS model, and dimension 2, which refers to collaborative learning, obtained very similar values, with respect to dimension 3, on the use of Socrative.

In the dimension "Cooperative Learning", a slight decrease in the median was observed (4.5), with a mean of 4.32 obtained. This indicates that the students positively valued cooperative learning as a strategy in the classroom. In addition, as in the previous dimension, the percentages with options 4 and 5 were the most repeated, with a total of 80.6% of the answers. Lastly, the dimension that referred to "Perspectives of the student on the VESS model" obtained the same mean as the dimension which referred to "Cooperative learning". The mean, although having a positive value, was the lowest obtained in the three dimensions studied (4.25). This trend, marked by the presence of worse scores in this dimension, was not observed in the percentages, as the most-repeated responses were 4 and 5 (81.9%).

*3.4. Correlational Analysis*

The correlational study of the three dimensions of the questionnaire was performed with Pearson's correlation test, with the results shown in Table 8.

**Table 8.** Results of the bivariate correlations of the items from the 3 dimensions of the questionnaire.

|  |  | VESS Model | Cooperative Learning | Socrative |
|---|---|---|---|---|
| **VESS Model** | R | 1 | 0.562 ** | 0.635 ** |
|  | P |  | 0.000 | 0.000 |
|  | N | 231 | 231 | 231 |
| **Cooperative learning** | R | 0.562 ** | 1 | 0.501 ** |
|  | P | 0.000 |  | 0.000 |
|  | N | 231 | 231 | 231 |
| **Socrative** | R | 0.635 ** | 0.501 ** | 1 |
|  | P | 0.000 | 0.000 |  |
|  | N | 231 | 231 | 231 |

**. The correlation is significant at 0.01 (two-tail). Source: Author created.

The results obtained showed the existence of a relationship between dimension 1 with dimension 2 and with dimension 3, as R = 0.562 and *p* = 0.000; and R = 0.635 and *p* = 0.000, respectively, with the given two-tailed significance at n.s. = 0.01. These values were moderate and high, respectively, as defined by [40]. Lastly, a relationship was also observed between dimension 1 and dimension 3, as R = 0.501 y *p* = 0.000, with the given two-tailed significance at n.s. = 0.01. In this case, the relationship was moderate, according to the authors cited previously.

## 4. Discussion and Conclusions

At present, we are experiencing radical changes in society in a short period of time. We have shifted from the era of knowledge to the era of intelligence. What is now important is not who owns knowledge, but who possesses the tools or criteria necessary for utilizing this knowledge—which is at everyone's reach in the best way possible—and the objective is to enable the correct decisions through thinking: critical thinking [41].

Teaching how to think, creating a culture of thinking, is one of the greatest challenges we have set for ourselves as teachers, so that a greater comprehension and autonomy of the students is achieved,

as well as significant learning. For this reason, we must make the students become aware of their thinking processes, build their own knowledge and create it in a reflective, critical and visible manner, as a group [42]. With respect to this, an instrument was created that could allow us to determine if the perception of the higher education students, as future teachers, on the use of the VESS model in the classroom, is positive, with the knowledge that its main axis is the fomenting of the culture of thinking in their students.

As previously pointed out, the validity of the instrument created to measure the perception of the students on the VESS model, cooperative learning to utilize the methodology in the classroom, as well as the use of Socrative as a digital tool for evaluation, is demonstrated.

The psychometric study of the scale proposed determined three factors, in both the EFA and the CFA, related with the VESS model (holism, thinking routines, externalization of thoughts and the mind). The second factor was related with cooperative learning, as referring to performance, understanding and personal values. Lastly, the third factor was related with the use of Socrative as a digital assessment tool, which allows for cooperative learning, increases motivation and the amount of learning, as well as the attitudes and skills related with critical thinking [43].

For the first dimension, the participants showed a good perspective towards the use of the VESS model, by considering education from the holistic point of view and granting value to the establishment of thinking as a resource for improving comprehension, and achieving critical and visible thinking. Ultimately, it improved metacognitive processes. In this sense, the results coincide with those obtained in previous studies [44], which sustain that when the students become involved in their own learning, it entails the creation of new expectations and an extra disposition towards the acquisition of knowledge. However, to directly involve the students in their learning, they should be motivated and their interest should be awoken. In this respect, ref. [45] defend the idea that pragmatic learning allows them to obtain ideas from the nearby surroundings they know, making them more creative.

The second dimension obtained in this study, related to cooperative learning, obtained a positive score by the participants, attesting that collaborative learning improves social interaction, cognitive skills, and promotes the construction of knowledge through critical thinking, as supported by [46].

As for the third and last dimension, on the use of Socrative, the results showed that this tool allows for adapting to the social context we live in, and interpreting the new realities, aside from promoting and applying critical-reflective thinking, propitiating this social transformation from the school as well, thereby allowing us to rethink the evaluation model that teachers must utilize in the midst of the 21st Century [47]. These data partially agree with those obtained in previous studies. Ref. [48] brings to light the clear preference of the students in training for active and novel methodologies that include "participative, individualizing, creative, and socializing styles" (p. 138).

In this sense, the mediatic alphabetization in general is going to create democratic and civic engagement. In addition, it is going to avoid the different inequalities which could exist at the social level, developing a reflexive and critical mind [49]. Furthermore, according to another study [50], the development of the ICTs are not just benefitting the student body and the society in general. They are affecting in a very direct way to the teaching role because it promotes the cooperative work and it allows work to be conducted that is more dynamic, motivating and adapted to the different learning rhythms. In conclusion, it allows work to be more inclusive.

Our study determines that the manner with which the teacher instructs the class only depends on the concept and concerns they have with respect to teaching, being in the hands of what Fried (1995), in [51] define as "the connection established between the passion for teaching and the quality of learning they want to obtain" (p. 27).

We can conclude that the questionnaire studied supports the applicability of this instrument for gathering the perception of the higher education student on the VESS model, as an effective and sustainable manner of teaching and learning.

Finally, it should be noted that this study has some limitations. On the one hand, the time allocated to data collection was limited because it depended on specific groups of students taking certain subjects,

so the sample was also one of the limitations. In addition, it should be noted that we only had degrees related to education, so in future research it would be interesting to apply this model to students from other fields. The few studies carried out on the VESS model and its repercussion on the teaching and learning processes could also be considered a limitation. This causes certain shortcomings in the bibliographic review carried out, as well as in the discussion of our study, as it is not possible to compare the results obtained in the different analyses with similar investigations.

**Author Contributions:** Conceptualization, M.H.R.-E. and M.D.H.-A.; formal analysis, J.M.M.-G.; investigation, J.M.M.-G., M.H.R.-E., and M.D.H.-A.; methodology, J.M.M.-G. and M.H.R.-E.; validation, J.M.M.-G.; writing—original draft, M.H.R.-E. and M.D.H.-A.; writing—review and editing, J.M.M.-G., M.H.R-E., and M.D.H.-A. All authors have read and agreed to the published version of the manuscript.

**Funding:** This research received no external funding.

**Conflicts of Interest:** The authors declare no conflict of interest.

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
