# Peer review of "Innovative Methodologies in a Pandemic: The VESS Model"

_sustainability, doi:10.3390/su12239952_

Round 1
Reviewer 1 Report
Please,
Tbl 1 (sample) could be more clearly arranged.
Tbl 3 pulled the number of lines of text into the factor structure.
The literature states the initial of the name and then the surname (although it should be reversed). Somewhere it is correct (the initial of the name and then the surname (15), and somewhere there is no name (33).
Please harmonize the citation of literature according to the propositions of the journal.
This reference has no function: B. S. Mózo, “Title 無 No Title No Title,” J. Chem. Inf. Model, vol. 53, no. 9, pp. 1689–1699, 2017, doi:
399 10.1017 / CBO9781107415324.004
The internal consistency of the factor items (tbl 6) can be inserted in tbl 2.
There is no need to repeat the content of tbl 8 in the text.
Author Response
Answers:
1. Table 1 has been refined, providing clearer information.
2. All errors related to citations and bibliographic references have been corrected.
3. It has been decided to leave the internal consistency values in table 6 to maintain the same structure in the EFA and CFA.
4. The wording related to table 8 has been simplified.
Reviewer 2 Report
Overall it is a good article with a solid basis and a topic of interest for Sustainability magazine. However, I recommend the following changes to the authors:
Figure 1 does not display well in the PDF document, it is cut off. You must add it so that it is correctly displayed.
The limitations of the study should be added in the discussion and conclusion sections.
The authors should add the following references, which are intrinsically linked to the subject of the article:
Forero-Carreño, F.A; Alemán de la Garza, L; Gómez-Zermeño, G. Teachers experiencies in ICT implementation at multigrade rural school. Edmetic. 2016, 5, 52-72.
Monreal-Guerrero, I; Parejo, J.L; Cortón de las Heras, M.O. Media literacy and the culture of participation: challenges for the digital citizen in the Information Society. Edmetic, 2017, 6, 149-167.
Author Response
Answers:
1. It has been decided to eliminate figure 1 because it is not relevant to the study.
2. The limitations of the research have been inserted in the discussion and conclusion section.
3. The bibliographic references recommended by the evaluator have been incorporated.